# Beta Blockers with Statins May Decrease All-Cause Mortality in Patients with Cardiovascular Diseases and Locally Advanced Unresectable Non-Small-Cell Lung Cancer after Chemoradiotherapy

**DOI:** 10.3390/cancers15041277

**Published:** 2023-02-17

**Authors:** Magdalena Zaborowska-Szmit, Sebastian Szmit, Marta Olszyna-Serementa, Paweł Badurak, Katarzyna Zajda, Anna Janowicz-Żebrowska, Aleksandra Piórek, Magdalena Knetki-Wróblewska, Piotr Jaśkiewicz, Adam Płużański, Maciej Krzakowski, Dariusz M. Kowalski

**Affiliations:** 1Department of Lung Cancer and Thoracic Tumors, Maria Sklodowska-Curie National Research Institute of Oncology, 02-781 Warsaw, Poland; 2Department of Cardio-Oncology, Centre of Postgraduate Medical Education, Institute of Hematology and Transfusion Medicine, 02-776 Warsaw, Poland; 3Clinic of Oncological Diagnostics and Cardio-Oncology, Maria Sklodowska-Curie National Research Institute of Oncology, 02-781 Warsaw, Poland

**Keywords:** lung cancer, mortality, cardiovascular diseases, beta blocker, statin, cardio-oncology

## Abstract

**Simple Summary:**

Concurrent platinum-based chemoradiotherapy (CRT) followed by maintenance treatment with the PD-L1 inhibitor durvalumab is the most effective therapy in unresectable stage III non-small-cell lung cancer (NSCLC). However, severe toxicity of this approach may lead to an unsatisfactory outcome. Cardiovascular diseases (CVD) may justify the use of sequential CRT to avoid severe adverse events and maintain satisfactory effectiveness. Ensuring that patients with CVD do not have a worse prognosis than patients without CVD is one of the goals of cardio-oncology. It is important that, after sequential CRT as personalized for patients with CVD, there is no increased mortality, and this is the first achievement of this study. Patients receiving beta-blockers and statins had lower all-cause mortality over 2, 3, and 4 years. The clear benefit of treatment with beta-blockers (in possible combination with statin) was confirmed in a subgroup of patients with CVD. The patients with CVD and indications for different cardiac therapy could live significantly longer if they received beta-blockers with or without statins during long-term follow-up. It may be a time to consider beta-blockers and statins as prevention strategy in patients undergoing CRT for NSCLC.

**Abstract:**

The study was conducted in the era when maintenance immunotherapy with durvalumab was not available in clinical practice after chemoradiotherapy (CRT) in unresectable non-small-cell lung cancer (NSCLC). The main aim of the study was to check whether the presence of cardiovascular diseases (CVD) and their pharmacotherapy affects the overall survival (OS) in such NSCLC patients undergoing sequential CRT. The group of 196 patients were analyzed: 101 patients with CVD (51.53%) and 95 patients with other reasons of qualification for sequential CRT (decreased performance status, older age, and other non-cardiovascular co-morbidities). Although patients with CVD were more often in older age, and they more often experienced cardiac and nephrological complications (*p* < 0.05 for all), there was a statistically nonsignificant trend for lower all-cause mortality in patients with CVD. The lowest all-cause mortality was observed in patients treated with beta-blockers and statins after two (HR = 0.31; 95%CI: 0.1–0.98; *p* = 0.047), three (HR = 0.33; 95%CI: 0.13–0.81; *p* = 0.015) and even four (HR = 0.45; 95%CI: 0.22–0.97; *p* = 0.027) years of follow-up. The benefit in OS remained significant in 101 patients with CVD treated with beta-blockers (HR = 0.65; 95%CI: 0.43–0.99; *p* = 0.045), and eventually statin, throughout the whole follow-up (log-rank *p* < 0.05). Further prospective studies are necessary to confirm the role of beta-blockers and statins in reduction of mortality in NSCLC patients undergoing radical CRT.

## 1. Introduction

A decrease in mortality due to cardiovascular reasons is the main goal of therapeutic intervention in cardiology. Similarly, in oncology, an array of therapeutic strategies leads to the prolongation of survival time due to the reduction of death caused by cancer disease. However, both specialties regard death for any reason as an important observation point. All-cause mortality seems to be crucial in cardio-oncology. Oncology patient treated with potentially cardiotoxic drugs may benefit with prolongation of progression-free survival or time to cancer disease recurrence, but toxicity of that treatment may also negatively impact the outcome. Damage of the heart due to anticancer therapy may lead to premature cardiac death and, in other cases, to quitting the cancer treatment and shortening of survival time. The biggest cardio-oncology registry, CARDIOTOX, shows a crucial relationship between diagnosis of cardiotoxicity and all-cause mortality (thereby not only in relation to death of cardiovascular cause), which means significantly shorter overall survival (OS) in oncology patients [1]. There are more examples where cardiotoxicity affects OS [2,3].

In locally advanced unresectable lung cancer, radiotherapy is part of radical treatment.Thus, the risk of premature death due to cardiological causes appears to be a disturbing effect of the therapy [4].An unfavorable survival time in a group of patients with higher doses of radiotherapy to the heart, when chemotherapy was not administered and in elderly patients, was observed [5]. It was proved that elderly patients may benefit from sequential chemoradiotherapy (CRT) [6]. This is probably due to the different biology of cancer in elderly patients and different risks of the appearance of distant metastasis [7].

The PACIFIC study has set a new standard for the treatment of locally advanced unresectable lung cancer. Looking at the results of 5-year survival after concurrent CRT, it can be concluded that the addition of maintenance immunotherapy with durvalumab increases the OS rate after this period from 33.4% to 42.9% [8]. It can be unequivocally stated that this is a valuable treatment with long-term benefits. However, not all patients from the real world could be qualified for treatment according to the criteria of the PACIFIC study [9]. One of the examples is a patient with concomitant heart disease. There are two cardio-oncological problems in this setting. Firstly, early and long-term complications (including cardiac complications) may cause an increase in mortality. Secondly, patients with coexisting cardiovascular diseases (CVD) should rather be considered for sequential CRT as the personalized and better tolerated treatment option, giving them a chance to later receive maintenance therapy with durvalumab. 

So far, the problem of long-term prognosis in patients with non-small-cell lung cancer (NSCLC) after CRT and concomitant CVD is poorly documented. The main hypothesis of the study is that sequential CRT is an adequate treatment option for patients with unresectable locally advanced NSCLC and CVD. The main aim of the study was to check whether the presence of concomitant cardiovascular diseases (CVD) and their pharmacotherapy affects the OS of patients with locally advanced unresectable NSCLC undergoing sequential CRT.

## 2. Methods

This study was a retrospective observation of patients qualified for sequential CRT in the period from July 2010 to October 2014. This means that the study was conducted in the era when maintenance immunotherapy was not available in clinical practice, but the therapeutic role of sequential CRT was assessed. The current available long-term follow-up can define prognostic determinants.

The inclusion criteria were defined as diagnosis of unresectable locally advanced NSCLC and the indication for sequential CRT due to significant comorbidities (including CVD) or older age or decreased performance status.

The exclusion criteria concerned patients who, nevertheless, qualified for concurrent CRT or received only palliative treatment (radiotherapy or chemotherapy) due to the advancement of NSCLC.

All patients were scheduled to receive 2–4 cycles of chemotherapy based on cisplatin or carboplatin. More than half of them received 2 full cycles of chemotherapy (106 patients; 54.1%), and every third received 3 cycles (65 patients; 33.2%). The preferred time to start radiation therapy was within the next cycle of chemotherapy, i.e.,3–4 weeks after the end of chemotherapy, or slightly longer in the case of slow bone marrow recovery. Administration of subsequent cycles of chemotherapy during or after radiotherapy was not recommended. The mean heart dose (MHD) during radiotherapy was median 12.36 Gy (interquartile range: 6.55–20.85 Gy).

Toxicity was determined according to Common Terminology Criteria for Adverse Events version 5. In the current study, attention was focused on complications that could affect the prognosis and to which the history of CVD could predispose. No complication had an immediate fatal outcome.

The primary point of observation was overall survival (OS), which was the time from the start of chemotherapy to the moment of death from any cause. 

The Kaplan–-Meier survival curves and Cox proportional hazards analysis were used to compare OS in patients with and without CVD. A special focus of this study was to assess the impact of treatment with different cardiac drugs on all-cause mortality. The oncological characteristic of the patients was compared in relationship with the presence of CVD and cardiac therapy by using the chi2 test with a possible Yates’ correction.

There was no detailed analysis of the dose and type of cardiac drugs. A special focus was associated with beta-blockers and statins, with which administration affected prognosis. Only the fact of their administration was taken into the account.

## 3. Results

The group of 196 patients were analyzed: 101 patients with CVD (51.53%) and 95 patients with other reasons of qualification for sequential CRT (decreased performance status, older age, and other non-cardiovascular co-morbidities). Among the 101 patients with history of CVD, the following diagnoses were present:➢arterial hypertension in 79 patients (40.3%),➢chronic coronary syndrome in 25 patients (12.8%),➢history of arterial thromboembolic events (ATE) in 32 patients (16.3%),➢history of venous thromboembolic events (VTE) in 7 patients (3.6%),➢atrial fibrillation in 4 patients (2.0%).

In addition to CVD, there were other significant prognostic internal diseases, such as chronic obstructive pulmonary disease (COPD) in 43 patients (21.9%), thyroid disease in 20 patients (10.2%), diabetes in 19 patients (9.7%), and lipid disorders in 12 patients (6.1%).

The detailed demographic and oncological characteristics of all 196 patients with NSCLC are presented in the Table 1. Patients with different diagnosis of CVD were significantly more often over the age of 65 years (*p* = 0.0003), significantly more often received chemotherapy without cisplatin (*p* = 0.009), and significantly more often experienced cardiac (*p* = 0.04) and nephrological complications (*p* = 0.03).

There was a statistically nonsignificant trend for lower all-cause mortality in patients with CVD: 44.55% vs. 53.68% (*p* = 0.2) after 2 years and 62.38% vs. 69.47% (*p* = 0.3) after 3 years (Figure 1). 

Patients treated with beta-blockers had significantly lower all-cause mortality (Figure 2) after two (HR = 0.57; 95%CI: 0.33–0.97; *p* = 0.039) and three (HR = 0.63; 95%CI: 0.41–0.98; *p* = 0.038) years of follow-up. 

Patients on beta-blockers had a better temporary outcome; however, they were less likely to have a good prognosis, which is defined as a baseline performance status KPS = 100 (Table 2). On the other site, patients treated with beta-blockers were less likely to experience deterioration of performance status, at least by 10 points in KPS, during CRT. Interestingly, patients treated with beta-blockers significantly more often had a longer time between the end of chemotherapy and the start of radiotherapy, which is defined as >42 days/6 weeks. No other significant differences were found in the oncological characteristics of NSCLC patients with a better prognosis due to beta-blocker administration.

The lowest all-cause mortality was observed in patients treated with beta-blockers and statins (Figure 3) after two (HR = 0.31; 95%CI: 0.1–0.98; *p* = 0.047), three (HR = 0.33; 95%CI: 0.13–0.81; *p* = 0.015) and even four (HR = 0.45; 95%CI: 0.22–0.97; *p* = 0.027) years of follow-up. The detailed results of the analysis of all-cause mortality risk are presented in Table 3.

Among the 47 patients receiving beta-blockers, as many as 46 were diagnosed with CVD, 18 had diagnosis of chronic coronary syndrome, 20 had a history of ATE, 35 had arterial hypertension, and one had atrial fibrillation. Of the 16 patients receiving beta- blockers and statin, all had CVD, 12 had chronic coronary syndrome and a history of ATE, and only 7 had arterial hypertension.

Analyzing the OS only in the subgroup of 101 patients with CVD, treatment with beta-blockers was still associated with a significantly better outcome (Figure 4). Thanks to beta-blocker therapy, all-cause mortality in CVD was reduced by 35% (HR = 0.65; 95%CI: 0.43–0.99; *p* = 0.045). Similarly, among patients with CVD, an improvement in OS was demonstrated for patients treated with a beta-blocker and statin (Figure 5). 

## 4. Discussion

### 4.1. Why Sequential Treatment?

In locally advanced NSCLC, for patients who are not eligible for surgery, chemotherapy followed by radiotherapy is a standard of care. The combination of chemotherapy and radiotherapy is more effective than radiotherapy alone [10,11]. In daily clinical practice, concomitant chemoradiotherapy may only be considered for patients in good or very good ECOG status, with normal respiratory function and no significant weight loss. In patients with comorbid respiratory or cardiovascular disease, the risk of concurrent CRT often outweighs the potential clinical benefit. If concurrent treatment is impossible for any reason, e.g., coexisting cardiovascular disease, treatment with sequential CRT becomes justified [12]. 

In the group of patients with coexisting cardiovascular diseases, it is reasonable to include a cardiological consultation in the process of qualification for CRT. The latest guidelines of the European Society of Cardiology recommend such consultation and cardiological follow-up after completion of anticancer treatment. In terms of risk stratification after radiotherapy alone, the mean heart dose becomes very important [13]. In addition, cytostatics used in lung cancer treatment may cause a number of cardiovascular complications [14]. The use of chemotherapy with cisplatin requires increased hydration, which may also affect the risk of deterioration of the cardiovascular system.

In the ESMO guidelines, the total and fractional dose of radiotherapy, the volume of the heart in the irradiated field, and the history of chemotherapy are the main risk factors for the cardiotoxicity of anticancer treatment. The authors did not discuss the possible increase in risk in patients treated concomitantly with chemoradiotherapy compared to sequential treatment [15,16]. It seems that in patients with contraindications to combination therapy, with particular emphasis on cardiovascular status, and in patients over 70 years of age, sequential combination of both methods should be preferred [17].

Analysis of the prognosis after CRT [18] shows that the diagnosis of reduced left ventricular ejection fraction (LVEF < 50%) was associated with nearly a twice as high mortality risk (HR = 1.97). In a retrospective, multicenter observation of patients with stage III NSCLC treated with CRT in the years 2006–2013 [19], out of 460 patients, 150 (32.6%) experienced cardiac complications, and the frequency of these complications was higher in patients with a cardiovascular history (44, 2%). The most common complications were arrhythmias (14.6%), heart failure (7.6%), and symptomatic coronary disease (6.8%). The significant independent risk factors were a history of cardiovascular disease (HR = 1.96, *p* < 0.01) and reduced baseline WHO fitness ≥ 2 (HR = 2.71, *p* < 0.01). When the risk in patients with a cardiac history was analyzed (138 patients, i.e., 30% of all analyzed patients), it turned out that 61 (i.e., 44.2%) patients developed a cardiac complication. The WHO-PS efficiency was assessed as ≥2, and chemotherapy with cisplatin were significant risk factors in both the univariate and multivariate analyses. The appearance of a cardiac complication did not significantly increase mortality, but it is worth noting that 5% of the patients died of cardiac causes (the percentage may be underestimated, because, in 17% of patients, the exact cause of death was not known). 

There are more and more studies showing that cardiovascular complications are quite common in patients after lung cancer treatment. The available studies showed that within 2 years after lung cancer treatment, the incidence of serious cardiac adverse events was 11–23% [20,21,22]. Comparing doses of standard radiotherapy [60 Gy] with high-dose conformal radiotherapy [74 Gy] with concomitant and consolidation chemotherapy, carboplatin + paclitaxel ± cetuximab in patients with stage IIIA and IIIB [23] showed worse overall survival (OS) for chemoradiotherapy with an increased dose to 74 Gy compared to the standard. This effect was related to the radiation dose to the heart. The safety of 127 patients with Stage III NSCLC with ECOG 0 to 1 who received escalating radiotherapy to 70–90 Gy (median, 74 Gy) was reviewed in six studies from 1996–2009 [4]. Symptomatic cardiac events (symptomatic pericardial effusion, acute coronary syndrome, pericarditis, significant arrhythmia, and heart failure) were the focus of observation. It appeared that 23% had one or more cardiac complications, with a median of 26 months to the first event (range, 1 to 84 months). The diagnoses included symptomatic pericardial effusion, myocardial infarction, unstable angina, pericarditis, significant arrhythmias, and heart failure. The rates of symptomatic cardiac events at 2 and 4 years were 14% and 32%. Patients with cardiac complications received higher doses of radiotherapy and were more likely to have associated coronary artery disease (35% vs. 8%) and a higher risk index according to WHO/ISH. In the univariate analysis, cardiac events were associated with, among others, radiotherapy dose and coronary artery disease (*p* < 0.001). Patients receiving a cardiac dose ≥ 20 Gy had a significantly higher risk of cardiac complications than those receiving <10 Gy (HR = 5.47; *p* < 0.001) or a dose in the range of 10–20 Gy (HR = 2.76; *p* = 0.03). Therefore, radiotherapy planning is crucial for oncological and cardiological prognosis. Our study confirmed that sequential CRT may be a personalized treatment option for patients with CVD, as we obtained a prognosis not worse than found in patients without CVD.

### 4.2. Why Beta-Blockers?

Radiotherapy can lead to various cardiovascular complications, especially in long-term observation [24]. Cardiac arrhythmias and autonomic dysfunction are common in short-term follow-up [25]. In inoperable advanced lung cancer, heart rate above 90/min can be understood as an unfavorable prognostic factor for overall survival [26]. Cardiac arrhythmias may also be associated with an unfavorable prognosis in lung cancer [27]. Beta-blockers are the basic drugs in the treatment of cardiac arrhythmias; hence, they are the likely reason for the improved prognosis in our observation and, not without significance, may give the effect of slowing down the heart rate.

Radiotherapy can also cause vascular complications, and in the treatment of symptomatic ischemic heart disease, beta-blockers have an established position. Therefore, it can be assumed that, also in the aspect of preventing cardiac ischemic events after radiotherapy, beta-blockers may play a beneficial role and thus reduce mortality in our population, especially in the period when the effect of radiotherapy on accelerating the heart rate and possibly inducing cardiac ischemia was observed.

The cardioprotective role of beta-blockers against heart damage caused by various oncological drugs is also being intensively analyzed. In the case of cardioprotection against anthracyclines, their importance is best documented [28]. However, blocking the sympathetic nervous system and inhibiting the effect of catecholamines on tissues may have its effects on cancer cells. It has been shown that if a cancer patient (especially breast cancer) receives a beta-blocker, she or he has a lower risk of death from any cause and death from cancer [29]. Another meta-analysis in patients with early-stage cancer (especially after surgery) showed a relationship between beta-blocker therapy and significantly longer OS and cancer-free survival (DFS) [30]. The beneficial effect of beta-blockers may be seen only in selected cancers [31,32]. 

In a large retrospective follow-up of NSCLC patients treated with radical radiotherapy, it was found that patients receiving beta-blockers had significantly longer distance metastasis-free survival (DMFS), disease-free survival (DFS), and overall survival [33]. Interestingly, in the multivariate analysis, this beneficial effect remained independent of the patients’ age, performance status according to Karnofsky, clinical advancement, histological type, concomitant chemotherapy, radiotherapy dose, tumor volume, coexisting hypertension, or chronic obstructive pulmonary disease. In another large retrospective analysis of NSCLC patients after radical radiotherapy, among cardiac drugs (ACEI, ARB, aspirin, and beta-blocker), only beta-blockers were associated with significantly longer DMFS, DFS, and OS [34]. 

Propranolol may prolong survival when using first line TKI-EGFR for lung adenocarcinoma [35]. Propranolol in combination with radiotherapy and cisplatin reduces the expression of kinase A(p-PKA), blocking the survival of clones of lung adenocarcinoma cells [36]. NSCLC patients receiving ICIs and beta-blockers had a better PFS [37]. A potential explanation is the observation that beta-blockers may increase the number of CD8+ CTLs [38].

### 4.3. Why Statins?

Statins not only lower blood cholesterol levels and reduce the risk of cardiovascular disease but may also play an important role in the prevention and treatment of lung cancer [39]. They have a number of anticancer properties, including the ability to reduce cell proliferation and angiogenesis, reduce invasion, and inhibit progression.

Several observational studies have investigated associations between statin use and lung cancer risk and lung cancer prognosis. In a follow-up of patients who received statins either before or after a diagnosis of lung cancer (11,051 and 3638 patients, respectively), it was noted that patients taking statins (especially simvastatin) had lower lung cancer-related mortality [40]. A large population-based study showed that if a statin was taken for dyslipidemia for at least 3 months before the diagnosis of lung cancer, mortality was significantly reduced [41]. A very significant reduction in mortality (HR = 0.58; *p* < 0.001) was obtained in patients with lung cancer treated with TKI-EGFR (additionally, also obtaining a significant prolongation of PFS) [42]. Among patients with stage IV NSCLC, 27% were taking statins and had significantly longer overall survival (HR = 0.76), as well as reduced lung cancer-related mortality. This effect was independent of the baseline oncological characteristics of the patients and the chemotherapy used [43].

In a meta-analysis of nearly 100,000 lung cancer patients, statin use was associated with a significant improvement in OS (HR = 0.79) and cancer-specific survival (HR = 0.83) [44]. Patients treated with statins after diagnosis of lung cancer (HR = 0.68) and patients with stage IV lung cancer (HR = 0.77) had a greater benefit in terms of OS. Another meta-analysis revealed that statins potentially potentiated the effects of tyrosine kinase inhibitors (HR = 0.86) and chemotherapy (HR = 0.86) on overall lung cancer survival [45]. 

Single studies showed the possible influence of statins on the prognosis of patients with locally advanced lung cancer. The retrospective analysis included 748 patients treated with chest radiotherapy, including 433 patients who underwent chemoradiotherapy [46]. In patients not taking statins, a mean heart dose ≥ 10 Gy was associated with a significantly increased risk of all-cause mortality (HR = 1.32; *p* = 0.022), while no such association was observed among patients receiving statins. Patients with locally advanced non-small-cell lung cancer have a high incidence of co-occurring coronary atherosclerosis and are at an additional cardiac risk after thoracic radiotherapy.

Another study showed a relationship between statin therapy and better regional control of stage III NSCLC in a 2-year follow-up after chemoradiation, but the result was not statistically significant [47]. Another study showed that the use of statins in NSCLC patients correlated with a higher BMI and predicted an improved survival in multivariate analysis (HR = 0.6) [48]. Obese patients, in this retrospective study, had significantly better survival compared to patients with normal weight.

The presented results indicate the need for further studies evaluating the impact of statin treatment on the survival of patients with locally advanced lung cancer undergoing chemoradiotherapy. It seems important to consider cardiovascular risk factors or the dose of radiotherapy to the heart, as well as the duration of therapy, drug compliance, and lipid assessment. Future research should also consider consolidation immunotherapy, as statins have been reported to have anticancer effects also through anti-inflammatory effects and immune regulation. In the literature, there are studies on the beneficial effects of statins in patients undergoing immunotherapy, which have been shown to be associated with a higher response rate and a longer time to treatment failure [49]. 

The present study has a number of significant limitations. This is a retrospective study, based on the experience of one center. Data were collected when maintenance immunotherapy after concurrent CRT had not yet been used. In addition, only patients undergoing sequential CRT as the treatment of choice for patients with CVD were included in the study. The effects of different beta-blockers and different statins have been evaluated, but dose-dependent effects and efficacy have not been confirmed. We have no data on what heart rate was achieved with a beta-blocker or how much cholesterol was lowered with a statin. It should be taken into account that the relationship between the development of cancer disease and heart failure can be very complicated [50,51]. The cardiotoxic effect itself may also correlate with cancer progression [52]. The role of angiotensin system inhibitors in neoplastic diseases is constantly discussed [53,54]. We also do not have a clearly defined antiplatelet and antithrombotic treatment for many cancers, which in lung cancer seems particularly important due to high prothrombotic readiness and the risk of arterial and venous thromboembolic complications [55]. Nevertheless, it is interesting to consider whether the inhibition of the sympathetic nervous system by beta-blockers and the protection of the vascular endothelium by statins with a possible anti-inflammatory effect actually can reduce all-cause mortality in lung cancer.

## 5. Conclusions

During sequential radical chemoradiotherapy, the prognosis of patients with concomitant cardiovascular diseases seems to not be less favorable. Patients taking beta-blockers for classic cardiological indications have a chance of significantly reduced all-cause mortality. When a beta-blocker was combined with a statin, all-cause mortality was reduced in all NSCLC patients with CVD during long-term follow-up. The results seem particularly relevant in terms of adding the possible maintenance immunotherapy with durvalumab.

## Figures and Tables

**Figure 1 cancers-15-01277-f001:**
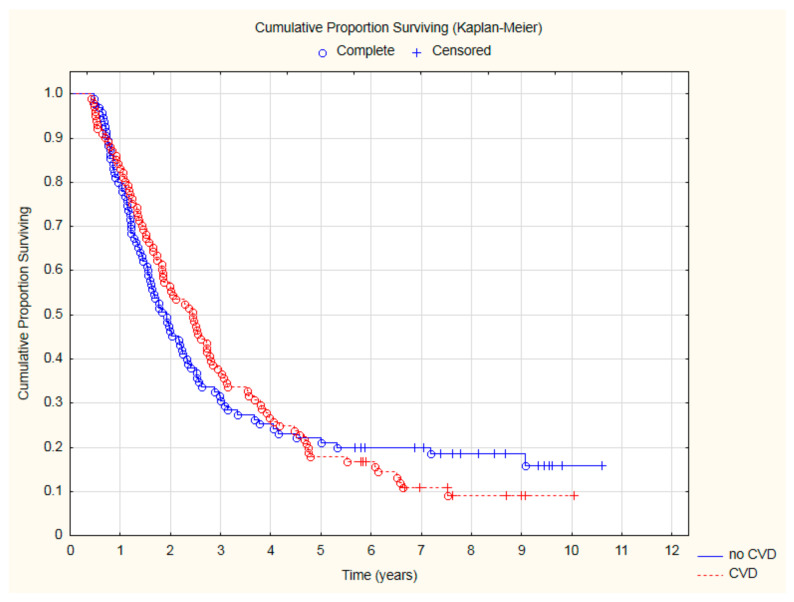
The comparison of overall survival in patients with and without CVD.

**Figure 2 cancers-15-01277-f002:**
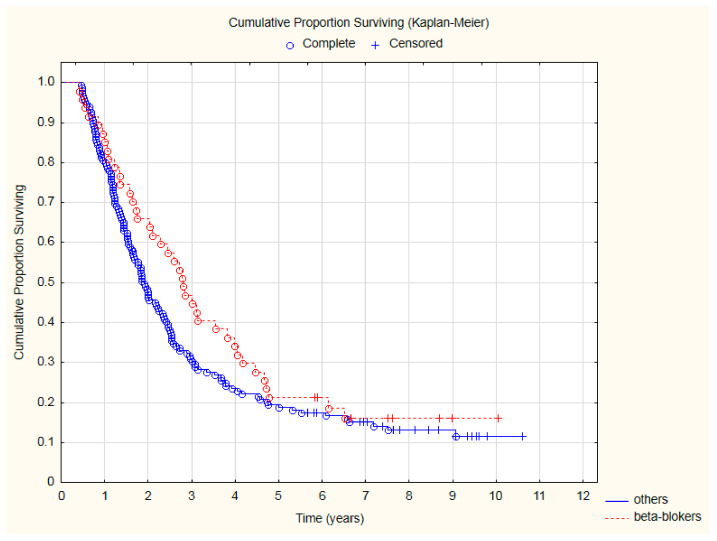
The comparison of overall survival in patients treated with beta-blockers due to CVD and other patients after CRT.

**Figure 3 cancers-15-01277-f003:**
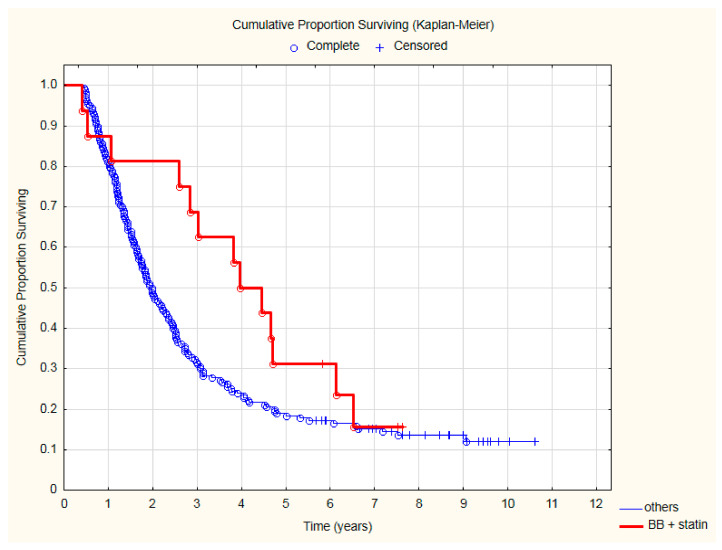
The comparison of overall survival in patients treated with beta-blocker (BB) and statin due to CVD and other patients after CRT.

**Figure 4 cancers-15-01277-f004:**
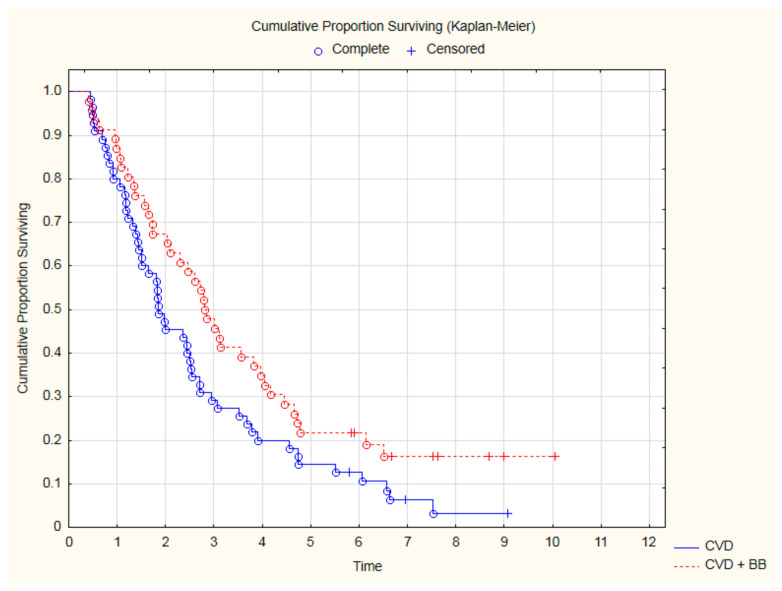
The comparison of overall survival among 101 CVD patients treated with beta-blockers (BB) or not after CRT (*p* = 0.04, log-rank).

**Figure 5 cancers-15-01277-f005:**
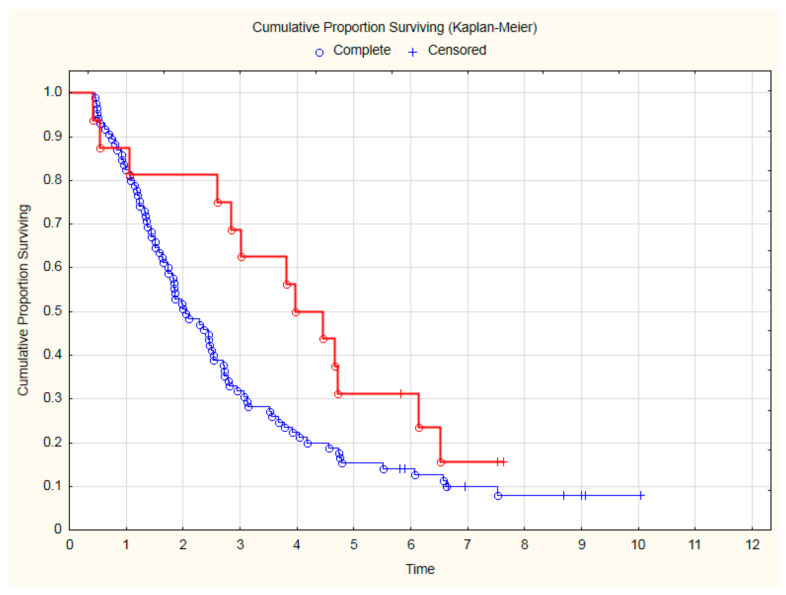
The overall survival among 101 CVD patients treated with beta-blockers and statin (red line) in comparison to patients without such therapy (blue line) after CRT (*p* = 0.035, log-rank).

**Table 1 cancers-15-01277-t001:** The characteristics of patients regarding presence of cardiovascular diseases (CVD).

		All 196 Patients	Presence of CVD		
			101 Patients with CVD	95 Patients without CVD	Chi-Square *p*-Value
Age > 65 years		52	38 (37.62%)	14 (14.74%)	0.0003
Clinical stage IIIA		94	52 (51.49%)	42 (44.21%)	0.31
Weight loss ≥ 10%		24	10 (9.9%)	14 (14.74%)	0.3
Baseline performance status KPS below 100		129	70 (69.31%)	59 (62.11%)	0.29
Deterioration of performance status at least by 10 points in KPS during CRT		53	23 (22.77%)	30 (31.58%)	0.17
Chemotherapy without cisplatin		21	17 (16.83%)	4 (4.21%)	0.009
Longer time between the end of chemotherapy and start of radiotherapy (defined as >42 days/6 weeks)		45	26 (25.74%)	19 (20%)	0.34
Complications all grades	Pneumotoxicity	24	11 (10.89%)	13 (13.68%)	0.55
	Nephrotoxicity	22	16 (15.84%)	6 (6.32%)	0.03
	Cardiac events	15	12 (11.88%)	3 (3.16%)	0.04
Complications grade 3/4 according to CTCAE	Febrile neutropenia	8	5 (4.95%)	3 (3.16%)	0.79
	Acute kidney injury	9	7 (6.93%)	2 (2.11%)	0.2
	Pneumotoxocity	5	3 (2.97%)	2 (2.11%)	0.94
	Cardiac events	5	5 (4.95%)	0	0.08
Response to CRT according to RECIST	CR	15	8 (7.92%)	7(7.37%)	0.82
	PR	123	66 (65.35%)	57 (60%)	
	SD	44	20 (19.8%)	24 (25.26%)	
	PD	14	7 (6.93%)	7 (7.37%)	
Type of cancer disease progression	Local progression	87	48 (47.52%)	39 (41.05%)	0.36
	Distant metastases	81	38 (37.62%)	43 (45.26%)	0.28
Subsequent chemotherapy		93	47 (46.53%)	46 (48.42%)	0.79
Secondary cancer disease		9	6 (5.94%)	3 (3.16%)	0.56

LEGEND: CVD-cardiovascular disease; KPS-Karnofsky Performance Status; CRT-chemoradiotherapy; CTCAE-Common Terminology Criteria for Adverse Events; RECIST-Response Evaluation Criteria in Solid Tumours; CR-complete response; PR-partial response; SD-stable disease; PD-progression disease.

**Table 2 cancers-15-01277-t002:** The comparison between patients on beta-blockers and patients without such therapy.

		Without BB (*n* = 149)	BB (*n* = 47)	*p*-Value
Age > 65 years		35(23.49%)	17(36.17%)	0.09
Clinical stage IIIA		68(45.64%)	26(55.32%)	0.25
Weight loss ≥ 10%		19(12.75%)	5(10.64%)	0.70
Baseline performance status KPS = 100		57(38.26%)	10(21.28%)	0.03
Deterioration of performance status at least by 10 points in KPS during CRT		46(30.87%)	7(14.89%)	0.03
Chemotherapy without cisplatin		15(10.07%)	6(12.77%)	0.60
Longer time between the end of chemotherapy and start of radiotherapy (defined as >42 days/6 weeks)		29(19.46%)	16(34.04%)	0.04
Complications all grades	Pneumotoxicity	21(14.09%)	3(6.38%)	0.16
	Nephrotoxicity	15(10.07%)	7(14.89%)	0.36
	Cardiac events	11(7.38%)	4(8.51%)	0.80
Complications grade 3/4 according to CTCAE	Febrile neutropenia	7(4.70%)	1(2.13%)	0.44
	Acute kidney injury	6(4.03%)	3(6.38%)	0.50
	Pneumotoxocity	4(2.68%)	1(2.13%)	0.83
	Cardiac events	3(2.01%)	2(4.26%)	0.40
Response to CRT according to RECIST	CR	13(8.72%)	2(4.26%)	0.31
	PR	92(61.74%)	31(65.96%)	0.60
	SD	32(21.48%)	12(25.53%)	0.56
	PD	12(8.05%)	2(4.26%)	0.38
Type of cancer disease progression	Local progression	66(44.30%)	21(44.68%)	0.96
	Distant metastases	60(40.27%)	21(44.68%)	0.59
Subsequent chemotherapy		70(46.98%)	23(48.94%)	0.81
Secondary cancer disease		7(4.70%)	2(4.26%)	0.90

LEGEND: BB-beta-blocker; KPS-Karnofsky Performance Status; CRT-chemoradiotherapy; CTCAE-Common Terminology Criteria for Adverse Events; RECIST-Response Evaluation Criteria in Solid Tumours; CR-complete response; PR-partial response; SD-stable disease; PD-progression disease.

**Table 3 cancers-15-01277-t003:** Therapies due to cardiovascular reasons and all-cause mortality.

Drugs	All-Cause Mortality				
	1 Year	2 Year	3 Year	4 Year	5 Year
	36(18.37%)	96(48.98%)	129(65.82%)	145(73.98%)	157(80.1%)
Beta-blocker (*n* = 47)	HR = 0.75 95%CI: 0.33–1.71 *p* = 0.492	HR = 0.57 95%CI:0.33–0.97 *p* = 0.039	HR = 0.63 95%CI:0.41–0.98 *p* = 0.038	HR = 0.70 95%CI:0.47–1.04 *p* = 0.074	HR = 0.79 95%CI: 0.54–1.14 *p* = 0.201
Statin (*n* = 26)	HR = 0.60 95%CI:0.18–1.95 *p* = 0.392	HR = 0.54 95%CI:0.26–1.12 *p* = 0.098	HR = 0.54 95%CI:0.30–0.98 *p* = 0.041	HR = 0.67 95%CI:0.41–1.12 *p* = 0.126	HR = 0.78 95%CI:0.49–1.24 *p* = 0.296
Beta-blocker + statin (*n* = 16)	HR = 0.67 95%CI:0.16–2.81 *p* = 0.587	HR = 0.31 95%CI:0.10–0.98 *p* = 0.047	HR = 0.33 95%CI:0.13–0.81 *p* = 0.015	HR = 0.45 95%CI:0.22–0.91 *p* = 0.027	HR = 0.56 95%CI:0.30–1.04 *p* = 0.067

## Data Availability

Data may be available upon reasonable request and with permission of the National Research Institute of Oncology in Warsaw (Poland).

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
