# Peer review of "Beta Blockers with Statins May Decrease All-Cause Mortality in Patients with Cardiovascular Diseases and Locally Advanced Unresectable Non-Small-Cell Lung Cancer after Chemoradiotherapy"

_cancers, 2023, doi:10.3390/cancers15041277_

Round 1
Reviewer 1 Report
Magdalena Zaborowska-Szmit et al show the manuscript entitled "Beta blockers with statins may decrease all-cause mortality in patients with cardiovascular diseases and locally advanced un- resectable non-small-cell lung cancer after chemoradiotherapy". A retrospective study on overall mortality in patients affected by CVDs and NSCLC treated by statin and beta-blockers.
Some comments:
- Methods. The Authors should specify more details on CRT of enrolled patients (i.e. timing, cycles).
- Results. The Authors should specify type of each class of complications. Type and timing of adverse events during the selected timeframe. Moreover, a detailed legend should be added to table 1. The same for table 2.
- Results. How many patients with BB were affected by CVDs? And how many patients treated with BB and statin had CVDs history? I would suggest to further assess the OS among CVD patients with and without BB and BB plus statin therapy.
- Results. What about other CV comorbidities? For example, hypertension, diabetes mellitus, hyperlipidemia? Could be helpful to add this info along the baseline characteristics.
- Discussion. I would suggest to discuss more about the relationship between CVDs and cancer. The Authors should add this ref according Santoro C, et al. Single, Dual, and Triple Antithrombotic Therapy in Cancer Patients with Coronary Artery Disease: Searching for Evidence and Personalized Approaches. Semin Thromb Hemost. 2021 Nov;47(8):950-961.
- A limitation section should be added to the manuscript.
Author Response
Magdalena Zaborowska-Szmit et al show the manuscript entitled "Beta blockers with statins may decrease all-cause mortality in patients with cardiovascular diseases and locally advanced un- resectable non-small-cell lung cancer after chemoradiotherapy". A retrospective study on overall mortality in patients affected by CVDs and NSCLC treated by statin and beta-blockers.
Some comments:
- Methods. The Authors should specify more details on CRT of enrolled patients (i.e. timing, cycles).
We have added the characteristics of CRT.
All patients were scheduled to receive 2-4 cycles of chemotherapy based on cisplatin or carboplatin. More than half of them received 2 full cycles of chemotherapy (106 patients; 54.1%), every third received 3 cycles (65 patients; 33.2%). The preferred time to start radiation therapy was within the next cycle of chemotherapy, i.e. 3-4 weeks after the end of chemotherapy, or slightly longer in the case of slow bone marrow recovery. Administration of subsequent cycles of chemotherapy during or after radiotherapy was not recommended. The mean heart dose (MHD) during radiotherapy was: median 12.36 Gy (interquartile range: 6.55 - 20.85 Gy).
- Results. The Authors should specify type of each class of complications. Type and timing of adverse events during the selected timeframe. Moreover, a detailed legend should be added to table 1. The same for table 2.
We have added the following explanation:
Toxicity was determined according to Common Terminology Criteria for Adverse Events version 5. In the current study, attention was focused on complications that could affect the prognosis and to which the history of CVD could predispose. No complication had an immediate fatal outcome.
We have added the LEGEND to tables.
LEGEND: CVD - cardiovascular disease; KPS - Karnofsky Performance Status; CRT - chemoradiotherapy; CTCAE - Common Terminology Criteria for Adverse Events; RECIST - Response Evaluation Criteria in Solid Tumours; CR - complete response; PR - partial response; SD - stable disease; PD - progression disease.
- Results. How many patients with BB were affected by CVDs? And how many patients treated with BB and statin had CVDs history? I would suggest to further assess the OS among CVD patients with and without BB and BB plus statin therapy.
We have added the following information:
Among 47 patients receiving beta blockers, as many as 46 were diagnosed with CVD, 18 had diagnosis of chronic coronary syndrome, 20 had a history of ATE, 35 had arterial hypertension, one had atrial fibrillation. Of the 16 patients receiving beta blockers and statin, all had CVD, 12 had chronic coronary syndrome and a history of ATE, only 7 had arterial hypertension.
We have performed analysis of OS among CVD patients.
- Results. What about other CV comorbidities? For example, hypertension, diabetes mellitus, hyperlipidemia? Could be helpful to add this info along the baseline characteristics.
We have added the following information:
The group of 196 patients were analyzed: 101 patients with CVD (51.53%) and 95 patients with other reasons of qualification for sequential CRT (decreased performance status, older age, other non-cardiovascular co-morbidities). Among 101 patients with history of CVD, the following diagnoses were present:
- arterial hypertension in 79 patients (40.3%),
- chronic coronary syndrome in 25 patients (12.8%),
- history of arterial thromboembolic events in 32 patients (16.3%),
- history of venous thromboembolic events in 7 patients (3.6%),
- atrial fibrillation in 4 patients (2.0%).
In addition to CVD, there were other significant prognostic internal diseases, such as chronic obstructive pulmonary disease (COPD) in 43 patients (21.9%), thyroid disease in 20 patients (10.2%), diabetes in 19 patients (9.7%) , lipid disorders in 12 patients (6.1%).
- Discussion. I would suggest to discuss more about the relationship between CVDs and cancer. The Authors should add this ref according Santoro C, et al. Single, Dual, and Triple Antithrombotic Therapy in Cancer Patients with Coronary Artery Disease: Searching for Evidence and Personalized Approaches. Semin Thromb Hemost. 2021 Nov;47(8):950-961.
We have extended the discussion.
- A limitation section should be added to the manuscript.
We have added the limitations of the study.
Reviewer 2 Report
Dear authors,
I have studied with interest the manuscript «Beta-blokers with statins may decrease all-cause mortality in patients with cardiovascular diseases and locally advanced unresectable non-small-cell lung cancer after chemoradiotherapy». The manuscript is well written and the work presented is original.
The results are clearly presented and all the conclusions are supported by the results. All the cited references are relevant to the research and well-balanced. The tables and figure correspond to the description in the text and they are well-designed and reflect important information.
However, I have comments that could improve the quality of the paper:
1. In the era immunotherapy and other perspective anticancer drug, CRT is not so widespread. Please, give better description of the necessity of such study exactely in the introduction, not only in the discussion.
2. Please, provide inclusion and exclusion criteria.
3. The main aim of the study was to evaluate all possible prognostic predictors, but the authors did not give what statistic methods were used to reveal those predictors. Please, provide information about covariates included in the analysis. The absence of this information did not allow making conclusion about the accuracy of the result.
4. Provide the structure of CVD. Definitely, the treatment with beta-blockers and statins were more beneficial for patients with CAD, which cannot be said, for example, about patients with congenital heart defects or heart failure of non-ischemic origin.
5. What was the dose of beta-blockers and statins and their names? Or only the fact of their administration was taken into the account.
Minor comment:
1. The primary point of observation was OS… Give the identification for this abbreviation.
2. The objective of the study should be replaced to “introduction”.
3. The main aim of the study was to evaluate all possible prognostic predictors…. Predictors of what? It should be clarify in the aim.
The authors have done considerable work and obtained very interesting data. However, the article requires significant revision before its publication.
Author Response
Dear authors,
I have studied with interest the manuscript «Beta-blokers with statins may decrease all-cause mortality in patients with cardiovascular diseases and locally advanced unresectable non-small-cell lung cancer after chemoradiotherapy». The manuscript is well written and the work presented is original.
The results are clearly presented and all the conclusions are supported by the results. All the cited references are relevant to the research and well-balanced. The tables and figure correspond to the description in the text and they are well-designed and reflect important information.
Thank You very much for your effort and willing to improve the quality of our study.
However, I have comments that could improve the quality of the paper:
- In the era immunotherapy and other perspective anticancer drug, CRT is not so widespread. Please, give better description of the necessity of such study exactely in the introduction, not only in the discussion.
We have added the following text to the INTRODUCTION:
The PACIFIC study has set a new standard for the treatment of locally advanced unresectable lung cancer. Looking at the results of 5-year survival after concurrent chemoradiotherapy, it can be concluded that the addition of maintenance immunotherapy with durvalumab increases the OS rate after this period from 33.4% to 42.9%. It can be unequivocally stated that this is a valuable treatment with long-term benefit. However, not all patients from the real world could be qualified for treatment according to the criteria of the PACIFC study. One of the examples is a patient with concomitant heart disease. There are two cardio-oncological problems in this setting. Firstly, that early and long-term complications (including cardiac complications) may cause an increase in mortality. Secondly, patients with coexisting cardiovascular diseases should rather be considered for sequential chemotherapy and radiotherapy as personalized and better tolerated treatment, giving them a chance to later receive maintenance therapy with durvalumab.
- Please, provide inclusion and exclusion criteria.
We have added clearly information about inclusion and exclusion criteria.
The inclusion criteria was defined as diagnosis of unresectable locally advanced NSCLC and the indication for sequential CRT due to significant comorbidities (including CVD) or older age or decreased performance status.
The exclusion criteria concerned patients who nevertheless qualified for concurrent CRT or received only palliative treatment (radiotherapy or chemotherapy) due to the advancement of NSCLC.
- The main aim of the study was to evaluate all possible prognostic predictors, but the authors did not give what statistic methods were used to reveal those predictors. Please, provide information about covariates included in the analysis. The absence of this information did not allow making conclusion about the accuracy of the result.
We have corrected the main aim of the study.
The main hypothesis of the study is that sequential CRT is an adequate treatment option for patients with unresectable locally advanced NSCLC and CVD.
The main aim of the study was to check whether the presence of concomitant cardiovascular diseases (CVD) and their pharmacotherapy affects the OS of patients with locally advanced unresectable NSCLC undergoing sequential CRT.
Please find the text about statistical methods:
The Kaplan-Meier survival curves and Cox proportional hazards analysis were used to compare OS in patients with and without CVD. Special focus of the study was to assess the impact of treatment with different cardiac drugs on all-cause mortality. The oncological characteristic of the patients was compared in relationships with presence of CVD and cardiac therapy by using the chi2 test with a possible Yates' correction.
- Provide the structure of CVD. Definitely, the treatment with beta-blockers and statins were more beneficial for patients with CAD, which cannot be said, for example, about patients with congenital heart defects or heart failure of non-ischemic origin.
We added the following supplementary information.
The group of 196 patients were analyzed: 101 patients with CVD (51.53%) and 95 patients with other reasons of qualification for sequential CRT (decreased performance status, older age, other non-cardiovascular co-morbidities). Among 101 patients with history of CVD, the following diagnoses were present:
- arterial hypertension in 79 patients (40.3%),
- chronic coronary syndrome in 25 patients (12.8%),
- history of arterial thromboembolic events in 32 patients (16.3%),
- history of venous thromboembolic events in 7 patients (3.6%),
- atrial fibrillation in 4 patients (2.0%).
- What was the dose of beta-blockers and statins and their names? Or only the fact of their administration was taken into the account.
We have added the following explanation.
There were no detailed analysis of the dose and type of cardiac drugs. Special focus was associated with betablokers and statins which administration affected prognosis. Only the fact of their administration was taken into the account.
Minor comment:
- The primary point of observation was OS… Give the identification for this abbreviation.
Please find our definition:
The primary point of observation was overall survival (OS), which was the time from the start of chemotherapy to the moment of death from any cause.
- The objective of the study should be replaced to “introduction”.
We have replaced the objective of the study to the INTRODUCTION.
- The main aim of the study was to evaluate all possible prognostic predictors…. Predictors of what? It should be clarify in the aim.
We have clarified the main aim of the study as:
The main aim of the study was to check whether the presence of concomitant cardiovascular diseases (CVD) and their pharmacotherapy affects the OS of patients with locally advanced unresectable NSCLC undergoing sequential CRT.
The authors have done considerable work and obtained very interesting data. However, the article requires significant revision before its publication.
Thank you once again for your help.
Round 2
Reviewer 1 Report
The Authors addressed all of my comments.
Reviewer 2 Report
The autors have adressed almost all my comments. But moderate English changes required, for example line 119 "....was associated with betablokers and statins which administration affected prognosis..." betablockers, not betablokers and so on.
And I did not find what predictors have been anilyzed by Cox proportional hazards analysis.